# Exploring the Potential of Pulse Transit Time as a Biomarker for Sleep Efficiency through a Comparison Analysis with Heart Rate and Heart Rate Variability

**DOI:** 10.3390/s23115112

**Published:** 2023-05-27

**Authors:** Jenna Bridges, Hossein Hamidi Shishavan, Adrian Salmon, Mark Metersky, Insoo Kim

**Affiliations:** 1Department of Medicine, University of Connecticut School of Medicine, Farmington, CT 06030, USAhamidi@uconn.edu (H.H.S.); 2Department of Biomedical Engineering, University of Connecticut, Storrs, CT 06269, USA; 3Division of Pulmonary, Critical Care and Sleep Medicine, University of Connecticut Health, Farmington, CT 06030, USA

**Keywords:** wearable sensors, continuous blood pressure, pulse transit time, heart rate variability, sleep dynamics, sleep efficiency, electrocardiography, photoplethysmography, polysomnography

## Abstract

The relationship between sleep dynamics and blood pressure (BP) changes is well established. Moreover, sleep efficiency and wakefulness during sleep (WASO) events have a significant impact on BP dipping. Despite this knowledge, there is limited research on the measurement of sleep dynamics and continuous blood pressure (CBP). This study aims to explore the relationship between sleep efficiency and cardiovascular function indicators such as pulse transit time (PTT), as a biomarker of CBP, and heart rate variability (HRV), measured using wearable sensors. The results of the study conducted on 20 participants at the UConn Health Sleep Disorders Center suggest a strong linear relationship between sleep efficiency and changes in PTT (r^2^ = 0.8515) and HRV during sleep (r^2^ = 5886). The findings of this study contribute to our understanding of the relationship between sleep dynamics, CBP, and cardiovascular health.

## 1. Introduction

Sleep is a vital aspect of human life, taking up a significant portion of our day, ranging from 20–40% [1]. It plays a crucial role in restoring energy, repairing tissues, and maintaining the normal functioning of the body. With adequate sleep, individuals can perform complex tasks with greater alertness and attention. However, disturbances in sleep patterns can have negative effects on both mental and physical health [2,3,4,5]. To accurately assess a patient’s sleep patterns and disturbances, specialized sleep doctors often prescribe a polysomnography (PSG) test. This in-patient overnight study measures various biological signals such as electroencephalogram (EEG), muscle movements, respiratory events, and heart rates (HR) to provide a comprehensive understanding of the patient’s sleep tendencies and characteristics. This controlled setting allows sleep doctors to confirm sleep diseases and disruptions.

In addition to traditional in-patient sleep studies, wearable devices can also be used to measure changes in the autonomic nervous system (ANS) during sleep, using various electrophysiological signals such as electrocardiography (ECG) and photoplethysmography (PPG). Numerous studies have linked changes in HR, especially heart rate variability (HRV), to changes in the ANS during sleep onset and arousal [6,7,8,9]. HRV is calculated using ECG-derived R-R peaks, and can be measured using time-domain and frequency-domain analysis [10]. The frequency domain of HRV is further broken down into three frequency bands: very-low-frequency [0.01–0.04 Hz], low-frequency (LF) [0.04–0.15 Hz], and high-frequency (HF) [0.15–0.40 Hz], which represent different autonomic variations and changes in HR and other cardiac functions including blood pressure (BP). During sleep, the sympathetic nervous system (SNS) and parasympathetic nervous system (PNS) activity fluctuate, causing similar fluctuations in the LF and HF components. The LF component is a marker of both sympathetic and parasympathetic activity, while the HF component is a quantitative indication of parasympathetic activity. The ratio between LF and HF values, known as the LF/HF ratio, reflects sympathovagal balance and the balance between LF and respiratory frequency power, both of which have an impact on HR control [6,8,10,11]. HRV parameters were also studied for their association with sleep quality [12]. Overall, HRV analysis is considered the gold standard for measuring changes in sleep dynamics during sleep.

BP is a crucial aspect of human physiology that fluctuates to maintain balance in the body, in coordination with the HR regulated by the ANS. BP varies over the 24 h sleep/wake cycle, being affected by daily activities, particularly during sleep. Over the course of one night’s sleep, BP drops by 10–20% compared to daytime levels for a healthy adult [13,14,15]. This drop, known as the BP dipping pattern, is due to increased parasympathetic activity and decreased sympathetic activity after sleep onset [14]. However, if the BP dipping during sleep is less than 10%, it is considered BP non-dipping, which is associated with a higher risk of cardiovascular problems [14,15]. BP non-dipping occurs when there are frequent arousals during sleep, and prevents the body from reaching each sleep stage fully. Prolonged BP non-dipping can lead to increased risk of hypertension, autonomic imbalance, metabolic dysfunction, and diabetes, due to increased sympathetic activity during the night [16,17]. Insufficient sleep quantity can lead to increased daytime BP over time, resulting in a higher risk of hypertension [18]. However, to date, continuous BP (CBP) changes during sleep or different sleep stages have not been fully understood.

The traditional method for measuring CBP is the invasive technique of arterial cannulation, which involves inserting a catheter into the radial artery [19]. Sphygmomanometer cuff measurement, a non-invasive method that uses occlusion to measure BP at intervals, is the standard for instantaneous BP measurement, but does not provide information about BP dipping [19,20]. Non-invasive CBP measurement methods using applanation tonometry have been explored to provide more accurate BP values by combining systolic and diastolic BP with arterial BP measurements [21,22,23,24,25,26]. However, these techniques are not suitable for sleep studies or continuous patient monitoring because they require manual handling by the examiner for accurate readings [18].

Pulse transit time (PTT) has also been examined as a potential non-invasive CBP biomarker and has been found to be inversely related to BP [20,27,28,29]. PTT refers to the amount of time it takes for a pulse pressure wave, originating from the heart, to reach another arterial site, such as the arm, wrist, finger, knee, or toe [30]. However, measuring CBP using PTT is challenging because it requires extensive calibration with respect to cuff-type or invasive BP monitors. Similarly, PPG alone [31] or in combination with other electrophysiological signals such as bio-impedance has been studied to estimate CBP values without extensive calibration [32]. Shahrbakaki et al. used PTT and HRV for scoring sleep staging [33]. Nonetheless, there are few studies investigated the relationship between CBP and sleep dynamics.

This study examined the relationship between sleep efficiency and PTT, as well as comparing it to the relationship between sleep efficiency and HRV. Given that the ANS synchronously controls HR and BP to maintain body homeostasis and the ANS activities can be measured through PTT and HRV without extensive calibration, we hypothesize that sleep efficiency will exhibit correlations with changes in PTT and HRV during the night. For this purpose, we recruited 20 adults among patients who had undergone PSG studies for diagnosing a mild sleep disorder. A custom-designed armband was used to simultaneously record PTT and HRV during the nightThe study analyzes the changes in PTT and HRV values at various points during the night and investigates their correlation with the sleep efficiency of each participant.

## 2. Materials and Methods

### 2.1. Custom Armband Design

We developed a custom-designed wearable armband. The armband consists of electronic circuits mounted on printed circuit boards (PCBs) measuring 25 × 27 mm^2^, and is equipped with a multi-LED PPG sensor, a 3-axis accelerometer, and a one-channel ECG, as depicted in Figure 1a. It is important to note that the 3-axis accelerometer was used for tracking participants’ activity levels during sleep. The two PCBs are stacked together and enclosed in a 3D-printed case measuring 30 × 30 × 5 mm^3^, which is then embedded in a thin, nylon-neoprene fabric secured with a hook and loop fastener, as shown in Figure 1c. This fastener can be adjusted to fit the arm circumference of any user, as the armband is worn on the upper arm. The armband also houses a 3.7 V rechargeable Li-Polymer battery in a pocket, which provides power to the electronics for the duration of a full night’s sleep.

An ADS1191 (Texas Instruments, Inc., Dallas, TX, USA) was used for the one-channel ECG. The ECG was sampled at a rate of 125 Hz. A MAX30101 (Maxim Integrated, San Jose, CA, USA) was used for the PPG, which was integrated with a multi-LED pulse oximeter sensor. The sensor was configured to utilize the built-in green LED (with a wavelength of 537 nm). The PPG was sampled at a rate of 100 Hz. A LIS2DH12 (STMicroelectronics N.V., Geneva, Switzerland) 3-axis MEMS accelerometer was used to measure activity. The wearable sensor unit was powered by a 1.8 VDC and 3.3 VDC power supply. The micro-USB connector attached to the board could be used to recharge the battery before and after use. A 32-bit microcontroller (STM32L433, STMicroelectronics, Switzerland) was used for synchronous data recording, which was stored in a micro-SD (secure digital) memory card. Both the micro-SD and the MCU were fixed on the PCB. The system’s functional view is illustrated in the block diagram in Figure 1b.

The location of the armband is crucial for obtaining clear signals and reducing noise. It is worn on the left arm of the human participants, with the photodetector of the PPG sensor situated above the left brachial artery. The PPG photodetector is housed within a 3D-printed enclosure, which not only protects it but also ensures consistent skin contact when the armband is worn properly. The ground and negative input ECG electrodes are attached to the left arm, while the positive input ECG electrode is attached to the chest. Figure 2 depicts the experimental setup with a participant of this study, who is wearing both the armband and PSG equipment.

### 2.2. Human Participants Study

A total of 20 participants were recruited for the study, including 11 males and 6 females. 17 datasets were used for the analysis, while 3 datasets were lost due to malfunctions with the armband or poor signal quality. The participants were between 29 and 69 years old, with a mean age of 49.4 ± 13.7 years, and represented diverse ethnic backgrounds including Hispanic, Caucasian, Asian, and African American. The study was conducted at the UConn Health Sleep Disorders Center within the UConn John Dempsey Hospital (Farmington, CT, USA). Participants were referred by Dr. Adrian Salmon, a board-certified sleep doctor, among patients who complained of poor sleep but were not diagnosed with a sleep disorder. To be eligible, participants had to meet the inclusion criteria of being 18 years or older and not suffering from insomnia. This was because the presence of insomnia could negatively impact the results of the study, and would not provide useful data for sleep dynamics measurement.

Participants in the study arrived at the UConn Health Sleep Disorders Center on the night of each scheduled PSG between 7:30 p.m. and 8:30 p.m. After providing informed consent and receiving necessary training for the armband, a sleep technician attached PSG equipment and the armband. The PSG began at “lights out”, at which point participants were instructed to go to bed and slept from approximately 10:00 p.m. to 6:00 a.m. The study ended when participants woke up or were awoken by the sleep technician, and the PSG sensors and the armband were turned off and removed.

The PSG test that UConn Health Sleep Disorders Center conducted includes several sensors: electromyogram, 6 differential channels of electroencephalogram, 2 channels of electrooculogram, and a 2-lead ECG. After the PSG recordings were taken during the night, they were sent to Dr. Adrian Salmon for scoring and interpretation. Based on the sleep study report, we calculated sleep efficiency, which is the ratio of total sleep time (TST) to total time in bed (TIB). We also extracted sleep onset time, final wakeup time, and the duration participants spent awake after sleep onset, known as WASO.

### 2.3. Signal Conditioning

To improve the quality of the ECG signal, a fifth-order bandpass filter with cutoff frequencies of 3–30 Hz was applied, which eliminates noise from local muscle activities and DC drift. The ECG signal was then processed using two artifact removal techniques, one to remove peaks not filtered by the bandpass filter, and another to eliminate ECG peaks with unrealistic physiological amplitudes. The PPG signal was also filtered using a fifth-order bandpass filter with cutoff frequencies of 1–6 Hz to remove noise from skin movement and the environment. Then, the armband signals were synchronized with PSG signals based on timestamps and the participant’s movements recorded by accelerometers.

### 2.4. Feature Extraction

#### 2.4.1. Inter-Beat Interval

ECG peaks were found using MATLAB 2021a (Mathworks, Inc., Natick, MA, USA), with a customizable distance between peaks and the prominence of those peaks. Then, the peak intervals were examined to remove any non-physiological data. The time between peaks was established as the Inter-Beat Interval (IBI), which was resampled for further heart rate (HR) parameter calculations. Then, IBI was segmented into 1 min time windows and a 1 min average IBI was calculated (IBI) (representing average HR).

#### 2.4.2. Heart Rate Variability

Next, HRV was calculated for time domain and frequency domain using the IBI data. For HRV TD analysis, IBI was also segmented into 1 min time windows; then, the root mean square of the standard deviation (hrvRMSSD) was calculated within each window. For HRV frequency domain analysis, the fast Fourier transform (FFT) spectrum was calculated using Welch’s periodogram method with 1 min time windows, from which the LF and HF bands were extracted. The area under each frequency band was then calculated and normalized. The parameters used for HRV frequency domain were the LF band (hrvLF) and HF band (hrvHF).

#### 2.4.3. Pulse Transit Time and Pulse Transit Time Variability

Calculating pulse transit time (PTT) followed the same process as finding IBI, except that peak detection was restricted to the window between two consecutive ECG R-peaks. The method was used to locate the systolic peak of the PPG signal within each defined ECG R-peak window, representing the peripheral peak detected by the PPG after an initial R-peak. The systolic peaks were also identified using MATLAB. PTT was calculated by dividing the 1 min windows into segments and finding the average PTT for each window (PTT). Outliers, representing PTT intervals that are not physiological for humans, were removed during filtering.

### 2.5. Analysis Method

Sleep efficiency was compared with five parameters: IBI, hrvRMSSD, hrvLF, hrvHF, and PTT. Changes in each parameter were analyzed at specific sleep points: the first 1 min epoch of sleep onset (SO), the maximum average 1 min epoch point during sleep (MaxP), the minimum average 1 min epoch point during sleep (MinP), and the first 1 min epoch of final wake up (WU). Figure 3 illustrates these four points of IBI, hrvHF, and PTT as an example. To better understand these changes, the difference and ratio between SO, MaxP, MinP, and WU were calculated for all five parameters and plotted against sleep efficiency. The coefficient of correlation (r^2^) was used to determine significance. A value of r^2^ greater than 0.5 indicates a strong correlation, a value smaller than 0.25 indicates a weak correlation, and a value between 0.25 and 0.5 is considered a moderate correlation. Additionally, correlations between sleep efficiency and several sleep time calculations from the validated PSG report were explored (see Figure 3).

## 3. Results

Table 1 presents a summary of the sleep efficiency parameters for 17 participants obtained from the PSG study. The participants showed a wide range of sleep efficiency values, ranging from 0.335 to 0.965. The study recorded multiple WASO events for all participants, of varying durations ranging from 10.5 to 256.5 min. Figure 4 demonstrates a strong positive correlation between sleep efficiency and the duration of WASO duration (r^2^ = 0.8536). However, no significant correlation was observed between sleep efficiency and the number of WASO events (r^2^ = 0.023).

The pattern of changes in HR during sleep reflects the level of arousal and the degree of physiological relaxation during different sleep stages. HR typically decreases during NREM sleep, and may become more variable and slightly increase during REM sleep. To better understand this relationship, this study compared IBI SO with IBI at MinP (fastest HR), MaxP (slowest HR), and WU. Figure 5 indicates a weak correlation between sleep efficiency and the difference between IBI at SO and at MinP (r^2^ = 0.0536), a weak positive correlation between sleep efficiency and the difference between IBI at SO and MaxP (r^2^ = 0.2236), and a weak correlation between sleep efficiency and the ratio of IBI at SO and WU (r^2^ = 0.2137).

Sleep is considered a condition in which vagal activity is high and sympathetic activity is relatively quiescent, which is considered true both for non-REM and REM sleep [34]. Therefore, hrvRMSSD will vary during sleep, and as a result, we do not expect any strong correlation between hrvRMSSD changes and sleep efficiency. As summarized in Table 2, sleep efficiency shows a weak correlation with the difference between WU and SO of hrvRMSSD, the difference between SO and MinP, and the ratio of SO and WU of hrvRMSSD.

In contrast, HRV frequency domain parameters have more correlation with sleep efficiency, since HRV is influenced by the ANS activity that undergoes significant changes during sleep. Phasic bursts of rapid eye movements during REM sleep may reflect sympathetic activation. Thus, hrvLF will have higher values, and hrvHF will have lower values during REM stages. As shown in Figure 6, the difference between SO and MinP of hrvHF has a moderate correlation (r^2^ = 0.4357) with sleep efficiency, whereas the difference between SO and MaxP of hrvLF exhibits a weak correlation (r^2^ = 0.014). It is worth noting that these correlations become strengthened when excluding two outliers (Participants 6 and 7). In particular, the difference between SO and MinP of hrvHF demonstrates a strong correlation (r^2^ = 0.5886) with sleep efficiency when excluding the outliers.

Figure 7 reveals that PTT changes have strong correlations with sleep efficiency for all cases. The difference between SO and MinP of PTT shows a correlation coefficient of 0.5208, while the difference between the MaxP and SO of PTT achieves a correlation coefficient of 0.6041 with sleep efficiency. The ratio of SO and WU of PTT showed the highest correlation coefficient of 0.8515 with sleep efficiency.

## 4. Discussion

This study aimed to investigate the relationship between sleep efficiency and both HRV and PTT in 20 patients who took a PSG study for sleep disorder diagnosis. As shown in Table 1, we obtained a wide range of sleep efficiency scores among the participants, which is unexpected. However, the wide range of sleep efficiency scores among participants provides diversity in the analysis, including both highly efficient and inefficient sleep in the same dataset. If the study only included healthy participants, it may not have resulted in the reported findings.

The study confirmed that sleep efficiency only correlated with WASO duration and not the number of WASO events. For example, Participants 1 and 7 in Table 1 had 14 and 16 WASO events, respectively, but different sleep efficiency scores of 0.909 (efficient sleep) and 0.335 (poorly efficient sleep). The reason for the difference is that Participant 1 has a total WASO duration of 19 min, while Participant 7 has a total WASO duration of 257 min. WASO is a marker that depicts sleep fragmentation, and moderate to highly fragmented sleep translates into poor sleep quality. Thus, both participants might suffer from low sleep quality. However, this study did not collect data on sleep quality; thus, exploring the links between WASO duration and sleep quality is beyond the scope of this paper and constitutes a limitation of this study.

PTT and IBI calculations are similar in utilizing the time between each R-peak, but are different because IBI uses the entire R-R interval time, while PTT uses the duration between each R-peak to PPG systolic peak time. By comparing both sides of each analysis (PTT and IBI), the interactions of both calculations with sleep efficiency can be better understood.

### 4.1. Sleep Efficiency Compared to IBI and PTT: SO-MinP

While there is no correlation between sleep efficiency and the difference between SO and MinP of IBI is (r^2^ = 0.0536), the difference between SO and MinP of PTT shows a negative strong correlation (r^2^ = 0.5208) with sleep efficiency. This means that participants with lower sleep efficiency have a more significant difference between SO and MinP of PTT, while participants with higher sleep efficiency have a smaller difference. The inverse relationship of PTT with BP estimation and the negative effect that WASO has on BP can be referred to. BP decreases during deeper sleep stages after sleep onset, but WASO events drastically increase BP due to an SNS response when a person wakes up. Longer WASO events keep BP at higher values during sleep. Therefore, MinP of PTT for a participant with a longer WASO duration is significantly lower than that for a participant with shorter WASO durations. In participants with higher sleep efficiency, the difference between SO and MinP of PTT decreases. As an example, Figure 3 shows the hypnogram along with IBI, HRV HF, and PTT of Participant 1 (SE = 90.9%). The value of PTT at SO is very close to that of the PTT at MinP.

### 4.2. Sleep Efficiency Compared to IBI and PTT: MaxP-Sleep Onset

The MaxP of both IBI and PTT may occur during deep sleep stages, because both HR and BP are typical lowest during deep sleep stages. Therefore, we anticipated that the correlation between sleep efficiency and MaxP of IBI and PTT would be similar. However, as shown in Figure 5b, the relationship between sleep efficiency and the difference between MaxP and SO of IBI is weak (r^2^ = 0.2236). In contrast, as shown in Figure 7b, the relationship between MaxP and SO of PTT and sleep efficiency shows a strong positive correlation (r^2^ = 0.6041). Figure 3 demonstrates that the participant had a long period of deep sleep when the PTT value showed the maximum value.

However, we found that IBI tended to be highly variable due to the shift in the power of the PNS and the SNS at sleep onset, making the measurement of IBI at SO inconsistent. As shown in Figure 3, IBI did not increase significantly at SO, and the MaxP appeared during light sleep, although IBIs are usually high during deep sleep stages. This inconsistency may be why little correlation between sleep efficiency and the difference between MaxP and SO of IBI is observed.

### 4.3. Sleep Efficiency Compared to IBI and PTT: Sleep Onset/Wake Up

Next, we compared the relationship between sleep efficiency and the ratio of SO and WU for both IBI and PTT. Due to the high fluctuation of IBI during the transition between consciousness and unconsciousness, the ratio of SO and WU of IBI cannot show any correlation with sleep efficiency. In contrast, the ratio of SO and WU of PTT may show a correlation with sleep efficiency if BP decreases during sleep, indicating restful sleep. This was confirmed by our results, as shown in Figure 5c and Figure 7c, wherein the ratio of PTT showed a strong correlation with sleep efficiency (r^2^ = 0.8515), while the ratio of IBI showed little correlation with sleep efficiency (r^2^ = 0.2137).

The PTT WU value was consistently higher than the PTT SO value for all participants. However, participants with low sleep efficiency had similar SO and WU of PTT ratios (1.1 or lower), while participants with higher sleep efficiency had significantly greater WU of PTT ratios compared to SO of PTT ratios (0.6 or higher). Next, we compared the relationship between sleep efficiency and the ratio of SO and WU for both IBI and PTT. Since IBI fluctuates highly during the transition between consciousness and unconsciousness, the ratio of SO and WU of IBI will not show any correlation with sleep efficiency. In contrast, the ratio of SO and WU of PTT will show a correlation with sleep efficiency if BP decreases during sleep, indicating restful sleep. This was confirmed by our results, as shown in Figure 5c and Figure 7c, wherein the ratio of PTT showed a strong correlation with sleep efficiency (r^2^ = 0.8515), while the ratio of IBI showed little correlation with sleep efficiency (r^2^ = 0.2137).

The PTT WU value was consistently higher than the PTT SO value. However, participants with low sleep efficiency had similar SO and WU of PTT ratios (1.1 or lower), while those with higher sleep efficiency had significantly greater WU of PTT ratios compared to SO of PTT ratios (0.6 or higher). Research suggests that BP dipping during sleep acts as a resetting mechanism to sustain a healthy level of BP during the day. When a person has non-dipping BP, the resetting of BP does not occur due to long durations of WASO throughout their night of sleep. This could lead to a decrease in the morning surge of BP, as the SNS tone has already dominated the PNS hours before waking up.

### 4.4. Sleep Efficiency Compared to HRV: Frequency Domain Analysis

Finally, we analyzed the relationship of the HF band for HRV with sleep efficiency. Previous research has shown that during sleep onset, HRV tends to have a higher HF tone and lower LF tone. Specifically, HF has been observed to dominate due to the high PNS tone during deep sleep stages, and HF values are at their lowest during REM episodes stimulated by high SNS tone. Our findings, as shown in Figure 3, confirm this observation, as the MaxP point appeared at a deep sleep stage, and the HF values remained consistently low during a long REM stage (around 05:20:00 in the figure).

Furthermore, we observed in Figure 6b that participants with lower sleep efficiency showed less of a difference between SO and MinP of hrvHF, indicating less of an increase in PNS tone at the initial sleep onset of hrvHF. For example, as shown in Figure 3, the PNS tone at sleep onset did not offset the SNS tone, which is necessary at sleep onset, resulting in the value at SO being closer to MaxP than that during REM or at MinP of hrvHF. Such a sustained SNS tone could lead to more frequent WASO episodes further into sleep, and could potentially supplement autonomic imbalances, a symptom consistent with hypertension [16].

Overall, the results of this study indicate that PTT and HRV HF may be reliable measures of sleep efficiency. In particular, the strong correlation between the PTT SO and WU ratio and sleep efficiency suggests that PTT could potentially be a useful tool in assessing an individual’s sleep quality and restfulness by reflecting BP changes during sleep. However, it should be noted that this study did not directly measure BP, and further research is needed to validate these findings and understand their underlying physiological mechanisms. A deeper understanding of the mechanisms underlying these measures could also lead to the development of new interventions to improve sleep quality and prevent sleep disorders.

## 5. Conclusions

In this study, we focused on investigating changes in PTT and HR during sleep to understand the differences between these values at sleep and awake, while observing these parameter changes as they relate to sleep efficiency. By examining these two parameters comparatively, we aimed to uncover parameters that illustrate sleep dynamics for measuring CBP trends. We analyzed sleep efficiency for linear relationships between PTT and HR calculations to further understand sleep dynamics using these values. Since PTT has an inverse linear relationship with continuous BP, many differences between PTT at sleep onset and other points during sleep that strongly correlate with sleep efficiency also show similar trends consistent with BP dipping and non-dipping in participants with low sleep efficiency. Although the correlation between the trends of PTT changes and BP changes during sleep cannot be validated due to the constraints of the study, we can appreciate the discovery of the consistent similarities between the tendencies of both parameters as they relate to frequent and prolonged WASO events.

In addition to the analysis of sleep efficiency, we found another strong linear relationship between sleep efficiency and the HF band of HRV for the variety of differences between sleep onset and the minimum point reached during sleep. Here, we discovered that the magnitude of difference between sleep onset and the minimum point reached during sleep correlates with the sleep dynamics of unrestful sleep (poor sleep efficiency), which could be due to WASO duration and not offsetting SNS tone in a deeper sleep. Overall, wearable sensor measurements were used to investigate PTT and HRV calculations with sleep efficiency. This analysis uncovers possible measurements of restful sleep in an individual through observing tendencies such as BP dipping and autonomic balance during different points of sleep.

## Figures and Tables

**Figure 1 sensors-23-05112-f001:**
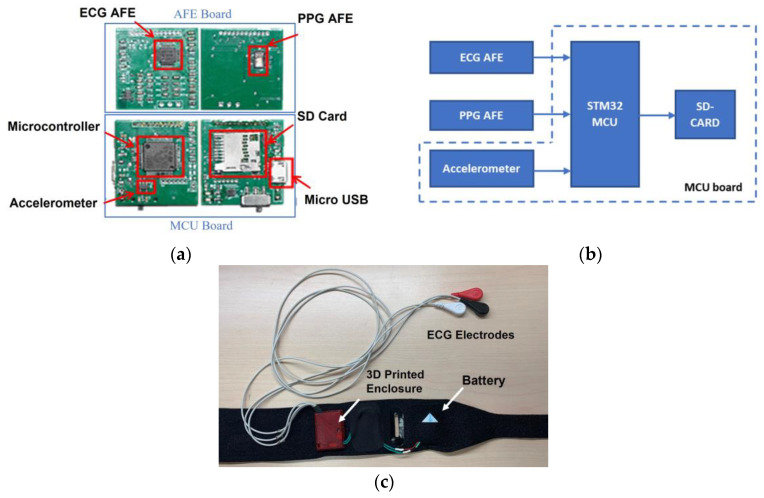
Custom-designed armband. (**a**) The front and back sides of the microcontroller (MCU) and the analog front-end (AFE) boards. (**b**) Simplified block diagram of the electronic boards. (**c**) Inner side of the armband, including electronics and ECG electrodes.

**Figure 2 sensors-23-05112-f002:**
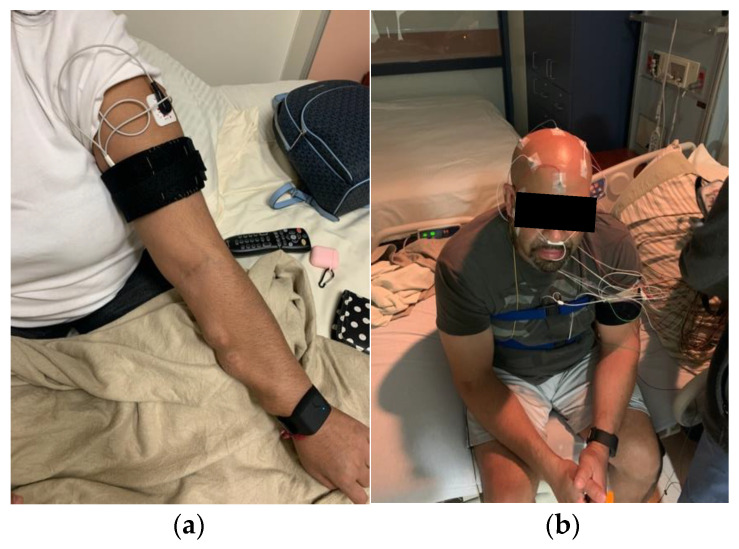
Experimental setup. (**a**) The armband worn on a participant’s upper arm. (**b**) A participant with PSG electrodes and the armband.

**Figure 3 sensors-23-05112-f003:**
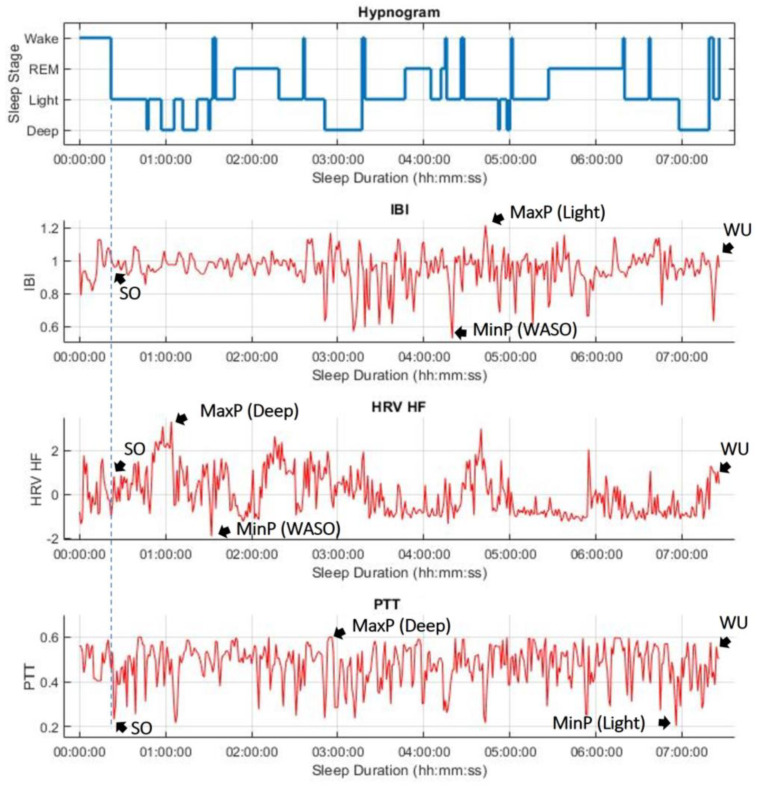
Hypnogram, IBI, HRV HF, and PTT averaged with 1 min epoch (Participant 1).

**Figure 4 sensors-23-05112-f004:**
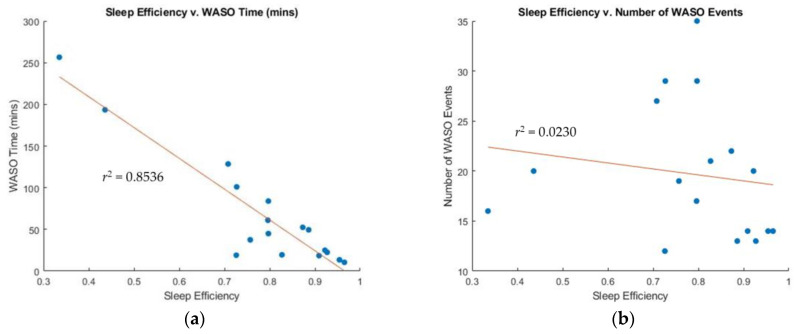
Correlation between sleep efficiency and WASO: (**a**) Sleep efficiency vs. WASO duration (mins), and (**b**) sleep efficiency vs. number of WASO events.

**Figure 5 sensors-23-05112-f005:**
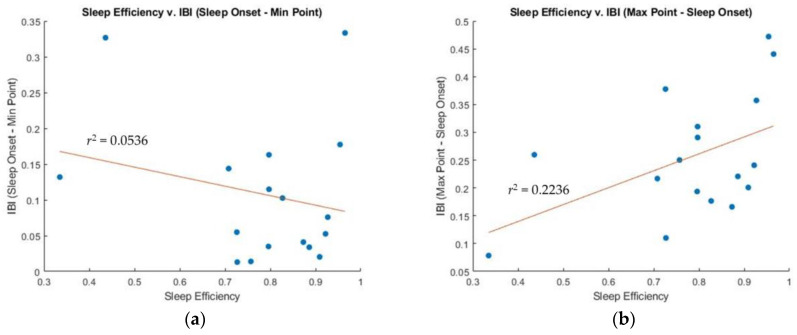
Correlation between sleep efficiency and the IBI. (**a**) Sleep efficiency versus IBI (SO-MinP), (**b**) Sleep efficiency versus IBI (MaxP-SO), and (**c**) Sleep efficiency versus IBI (SO/WU).

**Figure 6 sensors-23-05112-f006:**
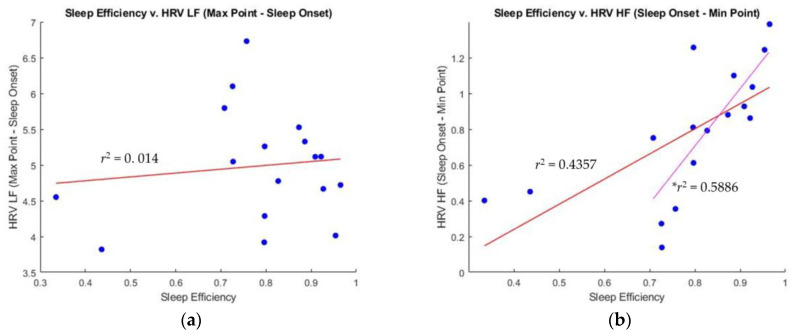
Correlation between sleep efficiency and hrvLF and hrvHF. (**a**) Sleep efficiency versus the difference between SO and MaxP of hrvLF. (**b**) Sleep efficiency versus the difference between SO and MinP of hrvHF (* r^2^ is the correlation coefficient when excluding outliers).

**Figure 7 sensors-23-05112-f007:**
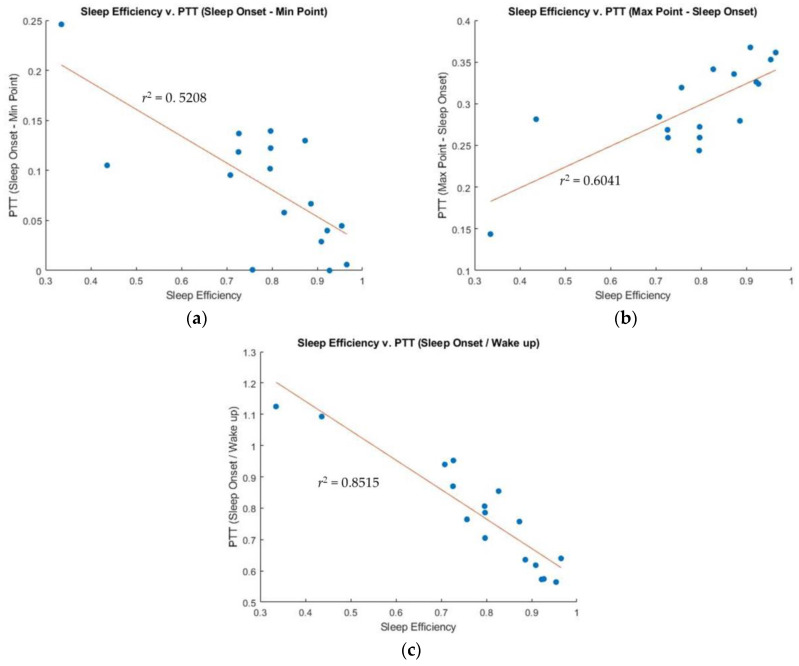
Correlation between sleep efficiency and PTT: (**a**) Sleep efficiency versus the difference between SO and MinP of PTT. (**b**) Sleep efficiency versus the difference of SO and MaxP of PTT. (**c**) Sleep efficiency versus the ratio of SO and WU of PTT.

**Table 1 sensors-23-05112-t001:** Sleep efficiency parameters obtained from PSG report.

Subject	SE	TIB (min)	TST (min)	TWT (min)	WASO (min)	Number of WASO
1	0.909	452	411	41	18.5	14
2	0.726	470.5	341.5	129	19	12
3	0.873	499.5	436	63.5	52.5	22
4	0.965	403.5	389.5	13.5	10.5	14
5	0.797	509	405.5	103.5	45	35
6	0.436	456.5	199	257.5	193.5	20
7	0.335	529.5	177.5	352	256.5	16
8	0.727	533	387.5	145	101	29
9	0.886	450	398.5	51.5	49.5	13
10	0.708	479.5	339.5	139.5	128.5	27
11	0.797	429	342	87	84	29
12	0.927	353.5	328	25.5	22.5	13
13	0.954	451	430	20.5	13.5	14
14	0.757	452.5	342.5	109.5	37.5	19
15	0.796	391	311	80	61	17
16	0.922	485	447	37.5	25	20
17	0.827	428.5	354	74	19.5	21

SE: sleep efficiency, TIB: time in bed, TST: total sleep time, TWT: total wake time, WASO: wakefulness after sleep onset.

**Table 2 sensors-23-05112-t002:** Summary of linear regression coefficients.

	IBI	hrvRMSSD	hrvLF	hrvHF	PTT
SO-MinP	0.0536	0.0256	0.0735	0.4357 (0.5886 *)	0.5208
MaxP-SO	0.2336	0.0945	0.014 (0.2177 *)	0.1113	0.6041
SO/WU	0.2137	0.0253	0.0843	0.0243	0.8515

* Correlation coefficient when excluding outliers (Participants 6 and 7) from analysis.

## Data Availability

The data presented in this study are available on request from the corresponding author. The data are not publicly available due to controlled unclassified information (CUI) under funding from the NIUVT.

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
