# Peer review of "Exploring the Potential of Pulse Transit Time as a Biomarker for Sleep Efficiency through a Comparison Analysis with Heart Rate and Heart Rate Variability"

_sensors, 2023, doi:10.3390/s23115112_

Round 1
Reviewer 1 Report
Congratulations for the article. I suggest you attach some suggestive bibliographic sources for your study:
- Shahrbabaki SS, Ahmed B, Penzel T, Cvetkovic D. Pulse transit time and heart rate variability in sleep staging. Annu Int Conf IEEE Eng Med Biol Soc. 2016 Aug;2016:3469-3472. doi: 10.1109/EMBC.2016.7591475. PMID: 28269047.
- Chalmers, T.; Hickey, B.A.; Newton, P.; Lin, C.-T.; Sibbritt, D.; McLachlan, C.S.; Clifton-Bligh, R.; Morley, J.W.; Lal, S. Associations between Sleep Quality and Heart Rate Variability: Implications for a Biological Model of Stress Detection Using Wearable Technology. Int. J. Environ. Res. Public Health 2022, 19, 5770. https://doi.org/10.3390/ijerph19095770
Author Response
Thank you for the review and comments. Please see attached.

Reviewer 2 Report
General comments:
The study aimed to explore the relationship between sleep efficiency and cardiovascular function indicators using warble sensors. The main hypothesis is that pulse transit time (PTT) is an adequate biomarker to track overnight changes in blood pressure (BP) without the need for calibration. However, the study failed to verify this hypothesis or provide any results due to the absence of BP measurements compared with PTT. Additionally, the method section should address the exclusion criteria for outliers, such as patients #6 and #7, in Table 2, as well as provide an explanation for why they were excluded. The lack of explanation about the outlier may negatively impact the validity and reliability of the study's results.
Specific comments:
1) Too many abbreviations interrupt reading the manuscript. Except for all measured indexes, recommend spelling out, such as VLF, SE, SO, WU, etc. Also, all abbreviations should address in tables and figure legends again. In addition, an abbreviation should be used for a word exactly, i.e., awakenings during sleep (WASO) à Wakefulness After Sleep Onset (WASO).
2) correlation coefficient should address in the figure instead of the figure legend.
3) What does "from the heart atria" mean? Is it an atrium or a heart?
4) Figure 7 may need to move to the Method section to help better to understand all measured indexes.
1) Change "It important" to "It is important" in line 102.
2) Change "were identified also" to "were also identified" in line 231.
3) Correct "week" to "weak" in line 280.
4) Delete duplicate words "during sleep" in line 304.
5) Change "correlation" to "a correlation" in line 308.
6) Change "in deeper sleep" to "in a deeper sleep" in line 473.
Author Response
Thank you for the comments. Please see attached.

Reviewer 3 Report
The article explores the relationship between sleep efficiency and cardiovascular function indicators such as pulse transit time (PTT) and heart rate variability (HRV) measured using wearable sensors. The study aims to understand the relationship between sleep dynamics, continuous blood pressure (CBP), and cardiovascular health. The article discusses the importance of sleep and the negative effects of disturbances in sleep patterns on both mental and physical health. The traditional method for measuring CBP is the invasive technique of arterial cannulation, while non-invasive methods using applanation tonometry have been explored to provide more accurate BP values. The pulse transit time (PTT) has also been examined as a potential non-invasive CBP biomarker and has been found to be inversely related to BP. The study hypothesizes that PTT is an adequate biomarker to track overnight changes in BP without the need for calibration. The article discusses the study conducted on 20 mild sleep-disordered patients to analyze the changes in PTT and HRV values at various points during the night and investigate their correlation with the SE of each participant. But I do have a few important questions and comments about the design and measurement of the control system.
Major comments:
(1) First of all, the subjects had to wear a variety of test devices, and they weren't pre-acclimated. So these devices may affect their sleep parameters to some extent. Whether individual movements and behaviors during sleep affect the data collected in the experiment.
(2) It was mentioned in the paper that all the subjects had certain sleep disorders, and no normal samples were detected. Is the experimental design needed to be further refined?
Minor comments:
(1) The name and order of Fig.3-Fig.6 must be the same.
(2) The orientation of the characters in Fig.1.(a) can be consistent.
(3) Label the part name in Fig.1.(C).
Author Response

(The authors gave the same response as above.)

Round 2
Reviewer 2 Report
Although the previous studies cited in the manuscript validated the relationship between PTT and BP, the current study has failed to demonstrate this correlation. In response, the author stated, “The aim of the existing studies is to estimate CBP using PTT.” However, there is no data available to estimate CBP using PTT. Furthermore, the conclusion section of the manuscript (lines 479-480) already acknowledged, “Although the correlation between the trends of PTT changes and BP changes during sleep cannot be validated due to the constraints of the study.” Therefore, it is recommended that the sentences in lines 89 - 92 be either removed or moved to the previous paragraph with appropriate modification. This will ensure that the manuscript accurately reflects the study's findings and does not make any unsupported claims.
